# Loss to Follow-Up from HIV Pre-Exposure Prophylaxis Care in Men Who Have Sex with Men in West Africa

**DOI:** 10.3390/v14112380

**Published:** 2022-10-28

**Authors:** August Eubanks, Bakary Coulibaly, Bintou Dembélé Keita, Camille Anoma, Ter Tiero Elias Dah, Ephrem Mensah, Gwenaëlle Maradan, Michel Bourrelly, Marion Mora, Lucas Riegel, Daniela Rojas Castro, Issifou Yaya, Bruno Spire, Christian Laurent, Luis Sagaon-Teyssier

**Affiliations:** 1Aix Marseille Univ, INSERM, IRD, SESSTIM, Sciences Economiques & Sociales de la Santé & Traitement de l’Information Médicale, ISSPAM, 13005 Marseille, France; 2ARCAD Santé PLUS, Bamako BP E2561, Mali; 3Espace Confiance, Abidjan 05 BP 1456, Côte d’Ivoire; 4Association African Solidarité, Ouagadougou 01 BP 2831, Burkina Faso; 5UFR Sciences de la Santé, Université de Ouahigouya, Ouahigouya 01 BP 346, Burkina Faso; 6Espoir Vie Togo, Lomé 7 BP 14543, Togo; 7ORS PACA, Observatoire Régional de la Santé Provence-Alpes-Côte d’Azur, 13005 Marseille, France; 8Coalition Plus, Community-Based Research Laboratory, 93500 Pantin, France; 9TransVIHMI, Univ Montpellier, INSERM, IRD, 34394 Montpellier, France

**Keywords:** HIV/AIDS, key populations, MSM, pre-exposure prophylaxis, PrEP, West Africa, community-based research

## Abstract

Loss to follow-up (LTFU) from HIV pre-exposure prophylaxis (PrEP) care compromises the goal of HIV elimination. We investigated the proportion of LTFU and associated risk factors among men who have sex with men (MSM) enrolled in a PrEP demonstration project in Burkina Faso, Côte d’Ivoire, Mali, and Togo. CohMSM-PrEP, a prospective cohort study, was conducted between November 2017 and June 2021 in community-based clinics. MSM aged 18 years or older at substantial risk of HIV infection received a comprehensive prevention package, including PrEP and peer education. LTFU was defined as not returning to the clinic for six months. Associated risk factors were investigated using a time-varying Cox’s model. Of 647 participants followed up for a median time of 15 months, 372 were LTFU (57.5%). LTFU was associated with younger age (adjusted hazard ratio [95% Confidence Interval]; 1.50 [1.17–1.94]), unemployment (1.33 [1.03–1.71]), depression (1.63 [1.12–2.38]), and perceiving no HIV risk with stable male partners (1.61 [1.23–2.10]). Contacting peer educators outside of scheduled visits was protective (0.74 [0.56–0.97]). Our findings show that LTFU from PrEP care in West African MSM is a major challenge to achieving HIV elimination, but that the involvement of peer educators in PrEP delivery helps to limit LTFU by providing users with adequate support.

## 1. Introduction

In West Africa, the HIV epidemic is concentrated in key populations, who, together with their sexual partners, accounted for 72% of all new HIV infections in 2020 [1]. Therefore, prevention in these populations is crucial to reach the global goal of eliminating HIV infection as a public health threat by 2030 [2,3]. Pre-exposure prophylaxis (PrEP) is an effective prevention tool [4,5,6,7,8,9]. PrEP programs in West Africa started very recently and are mainly for men who have sex with men (MSM) and sex workers.

The effectiveness of PrEP programs is highly dependent on its use by those targeted. Besides low access to PrEP, high rates of disengagement (i.e., discontinuation or loss to follow-up [LTFU]) from PrEP care constitute another major issue. Unfortunately, data on PrEP disengagement in Sub-Saharan Africa (SSA) is sparse due to slow roll-out rates [4]. This is notably the case for MSM in West Africa, specifically. This situation is attributable to the complicated legal and sociocultural position for MSM there. Indeed, discriminatory policies and laws against same-sex behaviors perpetuate a culture of widespread stigma and discrimination [10,11,12]. At the institutional level, political stakeholders relegate less financial resources to programs dedicated to these populations [13], limiting the availability of adapted HIV prevention and care for them, including PrEP provision [14,15,16,17,18,19]. This heteronormativity causes MSM to be rejected by society, leading to economic and psychosocial marginalization [10,20]. The effect of this hostile environment on PrEP disengagement is unclear, but will most surely have an impact.

Worldwide, rates of PrEP disengagement in MSM range from 17% to 58% [7,21,22,23,24]. Most data comes from the United States, while in SSA the literature is limited to a handful of studies in Kenya, South Africa, and Zimbabwe [25,26,27,28,29] and one in West Africa [30]. In a recent meta-analysis, including studies from all regions except SSA, the rate of both LTFU and PrEP discontinuation in MSM and transgender women was 29.3% [31]. From the rare studies in SSA, LTFU ranged from 19.1% in a PrEP program for MSM and sex workers [25] to 42.5% in MSM only, both in Kenya [27]. In general, the barriers that drive LTFU and discontinuation are similar to those that affect other steps of the PrEP care cascade [32,33,34]—A framework designed to identify gaps in PrEP implementation, spanning from initiation, to engagement (i.e., uptake and adherence), to disengagement [23,35,36,37,38,39,40,41,42]. Based on findings from MSM in the United States, discontinuation of PrEP most often occurred within the first months of taking it [32,43] and tended to worsen over time [43,44], while in Kenya, spending less time in care was a risk factor for LTFU [26,27]. The large majority of discontinuations were related to logistical or financial barriers to accessing PrEP services in both the US [45,46,47,48,49], and in Kenya [26,27]. Another major reason for discontinuation in the US was related to decreased HIV risk perception [45,47,48,49,50,51,52,53], due to abstinence or entering into a monogamous relationship [54]. Younger age was associated with LTFU in the US only [45,47,53,55,56,57], while, in both SSA and in the US, substance use [26,27,46], PrEP side effects [28,29,45,47,50,52,58] and psychosocial problems [51,53], including stigma [25,28] were risk factors for disengagement.

One of the select few demonstration projects to provide PrEP to MSM in West Africa was CohMSM-PrEP [59]. From November 2017 until June 2021, MSM participants had access to PrEP and peer education as a part of a comprehensive sexual health prevention package in community-based clinics [30]. Initial findings showed that PrEP uptake helped prevent new HIV infections [30], and that certain community-based aspects of the cohort and accurate HIV-risk perception facilitated PrEP initiation and engagement [60,61]. Further findings showed that some factors related to the region’s hostile environment, like socioeconomic and psychosocial vulnerability were barriers to PrEP engagement and effective HIV protection [60,62]. Finally, preliminary data suggested that LTFU was high with 27% of participants being LTFU at the time of analysis.

In the present study, we aimed to estimate the proportion of participants LTFU at the cohort’s conclusion and to identify the risk factors for LTFU in CohMSM-PrEP. Our main hypothesis was that similar barriers and facilitators influencing the other steps of the PrEP care cascade would also play a role in study retention. To our knowledge—beyond the preliminary data from CohMSM-PrEP—There have been no other studies on retention of MSM in PrEP care in the West African context. Furthermore, this is the first study to explore risk factors for LTFU among MSM taking PrEP in West Africa and one of the very few in SSA. Informing on the state of the PrEP care cascade in West Africa and identifying a LTFU profile, will help guide PrEP delivery and rollout as more and more countries in the region add PrEP to their national AIDS programs.

## 2. Materials and Methods

### 2.1. Study Design and Participants

CohMSM-PrEP was a prospective cohort study initiated on 20 November 2017 and ended on 30 June 2021. It assessed the acceptability and feasibility of a comprehensive sexual health prevention package, including PrEP, for MSM in community-based clinics in West Africa. The four study sites were MSM-friendly clinics run by community-based organizations: Centre Oasis run by the Association African Solidarité (AAS) in Ouagadougou (Burkina Faso); Clinique de Confiance, run by the association Espace Confiance in Abidjan (Côte d’Ivoire); Clinique de Santé Sexuelle des Halles run by the Association pour la Résilience des Communautés pour l’Accès au Développement et à la Santé—ARCAD Santé PLUS (formerly ARCAD-SIDA) in Bamako (Mali); and Centre Lucia, run by Espoir Vie Togo (EVT) in Lomé (Togo). CohMSM-PrEP was designed as an addition to, and development of, a previous MSM cohort, CohMSM, which studied the implementation of HIV prevention and care services for MSM in the same study sites [63]. Except for PrEP provision, the comprehensive sexual health prevention package was the same in both cohort studies.

First, participants from CohMSM who wished to continue follow-up and take PrEP were recruited, thereby ending their participation in CohMSM. Then, “new” participants were identified by peer educators (PE) through a specific network of community-based organizations. Both ex-CohMSM participants and “new” participants had to meet the same eligibility criteria (a comparison of the two cohorts has been previously described [61].

Participants were eligible if they were 18 years or older, HIV-negative (status confirmed at study enrollment), MSM (defined as reporting at least one episode of anal intercourse [insertive or receptive] with another man in the six months preceding enrollment), and reported any of the following HIV at-risk criteria: (i) Non-virally suppressed seropositive sexual partner (male or female), (ii) condomless anal or vaginal sex with two or more partners in the previous six months, (iii) a history of sexually transmitted infection (STI) in the previous six months, (iv) post-exposure prophylaxis use in the previous six months, or (v) requesting PrEP. Exclusion criteria included signs or symptoms of acute HIV infection, probable exposure to HIV, a creatinine clearance of less than 60 mL/min (using the Cockroft-Gault equation); positive or indeterminate hepatitis B surface antigen test; and allergy or contraindication to PrEP drugs.

### 2.2. Procedures

Generic fixed-dose PrEP combinations of tenofovir disoproxil fumarate 300 mg and emtricitabine 200 mg were prescribed as follows: daily (one pill per day) or event-driven (2 + 1 + 1 dosing; i.e., 2 pills between 2–24 h before sex [1 if PrEP taken the previous day] followed by 1 pill 24 h and another 48 h after the first pill[s]) [64,65]. At each quarterly follow-up visit, in concertation with study doctors and/or PE, participants could decide to switch PrEP strategies or stop PrEP (temporarily or permanently) depending on their needs. The study procedures were the same regardless of the PrEP regimen.

During quarterly follow-up visits, participants received a refill of their PrEP prescription, clinical examinations by a physician, HIV testing, screening and treatment for other STIs, condoms and lubricants and peer-led psychosocial support and counseling. The latter included guidance on adherence, condom use, switching between PrEP strategies, abstaining from certain high-risk activities and study retention. Adherence-specific counseling was provided monthly beginning with first PrEP delivery and every three months thereafter. Furthermore, participants could come to their clinic for an unscheduled visit or contact PE by telephone at any time. If participants were 15 days late for their scheduled visit, PE contacted them by telephone (with previous consent). All services were offered free of charge and participants were compensated USD 5 for transport costs at every scheduled visit.

Participants were screened for HIV using national algorithms (Abbott Determine HIV 1/2 assay [Abbott Laboratories, Chiba, Japan] and, if the result was positive, SD Bioline HIV-1/2 3.0 [SD, Gyeonggi-do, South Korea] or First Response HIV-1/2 assay [Premier Medical Corporation, Mumbai, India]) and those diagnosed HIV positive during follow-up were invited to initiate antiretroviral treatment immediately.

Trained research assistants administered standardized face-to-face questionnaires at enrollment and every three months thereafter to collect socio-demographic and behavioral data on individual characteristics, cohort and PrEP-related factors, MSM-identity, psychosocial aspects, sexual behaviors, and substance use. Medical staff collected clinical data at each follow-up visit (scheduled or not), including LTFU, PrEP regimen and HIV and STI testing results.

### 2.3. Measures

#### 2.3.1. Event of Interest (Outcome)

In the present study, the event of interest was the first episode of LTFU. A participant was defined as LTFU if they did not return to the clinic six months after their last visit, regardless if they re-engaged later in care. Participants refusing to continue in the cohort or those asking to withdraw their consent were also defined as LTFU. The outcome is time to LTFU. It was constructed as the time (in months) elapsed between the date of enrollment in the cohort (first prescription of PrEP) and the date of the last observed visit, before an absence of more than six months. Data for participants that completed follow-up were censored at the last quarterly visit before the end of the study (i.e., 30 June 2021). For HIV seroconverted participants, data was censored at the date of their positive HIV test; and for deceased participants, data was censored at the date of their death. A dichotomous variable indicated whether data for participants was censored (=1) or not (=0). Examples to illustrate the definition of LTFU in our study can be found in Appendix A.

#### 2.3.2. Covariates

Covariates in the present analysis included:

Sociodemographic and socioeconomic characteristics. Age (grouped, 18–24 vs. >24 years old), country-fixed effects (Burkina Faso, Côte d’Ivoire, Mali, and Togo), employment status (employed vs. unemployed).

Cohort or PrEP-related characteristics. Recruitment type (ex-CohMSM participant vs. new participant), chosen PrEP regimen (event-driven or daily), self-reported PrEP adherence during most recent sexual intercourse (daily regimen, ‘optimal’ [7 pills taken in the week before most recent intercourse], vs. ‘suboptimal’ [4–6 pills], vs. ‘poor’ [1–3 pills], vs. ‘no PrEP’ [no pills]; event-driven regimen, ‘optimal’ [dosing schedule respected completely], vs. ‘suboptimal’ [taking at least one pill 2–24 h before sex act and one pill 24 h after sex act], vs. ‘poor’ [all other pill taking combinations], vs. ‘no PrEP’ [no tablets taken before or after sex act] [30]), contacted a PE outside of scheduled visits (‘yes’ vs. ‘no’).

MSM identity and psychosocial aspects. Self-defined sexual orientation (‘heterosexual’,‘homosexual/gay’ or ‘bisexual’), self-defined gender identity (‘man/boy’ vs. ‘both a man and a woman’, ‘more a woman’, and ‘neither a woman nor a man’), depression (based on the Patient Health Questionnaire-9, where a minimal (moderate to high) depression score was defined as ≤4 (>4) [66].

Sexual behaviors. Have a casual male/female partner (‘yes’ vs. ‘no’), number of male partners in the previous three months (≤2 vs. >2), HIV risk perception score with stable/casual partners ranged from 0–10 and was defined as no risk = 0, at risk = 1–10, and no stable/casual partner, combined prevention use during most recent anal intercourse (‘no PrEP & no condom use’ vs. ‘only PrEP use’ vs. ‘condom use only’ vs. ‘PrEP & condom use’).

### 2.4. Statistical Analysis

First, participants’ characteristics at baseline were described. In all longitudinal analyses, which ranged from baseline to M42 and used repeated measures, country and recruitment type were considered to be time-constant (i.e., measured at baseline); while the remaining covariates were time-dependent (i.e., varying over time). Next, bivariate analysis was conducted using the Kaplan–Meier technique to describe and explore significant associations between time-to-LTFU and covariates. Then, a time-varying Cox’s proportional hazards model was used to assess risk factors for LTFU. We identified potential covariates for the multivariate model using a threshold of 20% significance in univariate analysis. All univariate and multivariate models were adjusted for country-fixed effects. The forward selection technique was used to construct the final multivariate model and its goodness-of-fit was verified using the Akaike information criterion (AIC). All analyses were performed using STATA 16.1 statistical software.

### 2.5. Ethical Considerations

CohMSM-PrEP was registered with ClinicalTrials.gov, number NCT03459157. The study protocol was approved by the ethics committees of Mali (N°2017/113/CE/FMPOS), Burkina Faso (N°2017-7-105), Côte d’Ivoire (N°088/MSHP/CNER-kp) and Togo (N°338/2017/MSPS/CAB/SG/DGAS/DPML/CBRS), and Belgium (ethics committee of the Antwerp University). All participants provided written informed consent.

## 3. Results

### 3.1. Sample Characteristics

From 20 November 2020 until 30 June 2021, 647 participants enrolled in the CohMSM-PrEP study. Table 1 describes the characteristics of the study sample at baseline. Mean age was 25.3 years old (standard deviation, SD = 5.8) and 48.5% of participants were 18–24 years old. A majority of participants came from Mali (39.7%), 20.3% from both Côte d’Ivoire and Togo, and 19.7% from Burkina Faso. Almost half (46.2%) of participants were unemployed.

Half of the study sample (49.8%) were ex-CohMSM participants. A forth (25.0%) of participants contacted PE outside of their scheduled visits. Three-fourths (73.1%) of participants chose event-driven PrEP at baseline and after three months of pill intake, a third (40.9%) had optimal PrEP adherence, 16.1% had suboptimal adherence, 5.2% had poor adherence and 21.3% did not take PrEP during the week preceding their most recent intercourse.

At baseline, a majority of the study sample self-defined as bisexual (51.6%), 35.2% as homosexual or gay, and 2.3% as heterosexual. Over half (54.7%) of participants self-identified as a man or boy. Forty-five percent of the study sample (45.6%) had a moderate to high depression score (PHQ-9).

In the previous three months, 58.6% of the study sample had casual male partners, 20.3% had casual female partners and over a third (34.6%) of participants had more than two male sexual partners. With casual partners, 43.4% perceived themselves to be at risk of HIV infection, 14.8% at no risk, while the rest (40.2%) were not concerned because they had no casual partners. With stable partners, 41.0% perceived themselves to be at risk of HIV infection, 29.7% at no risk, while the rest (28.0%) were not concerned because they had no stable partner. After three months of pill intake, and in terms of combined prevention use during most recent intercourse, 33.0% used both PrEP and condoms, 9.6% used condoms only, 28.6% used PrEP only, and 26.7% used neither PrEP no condoms.

### 3.2. Loss to Follow-Up and Risk Factors

Overall, the median follow-up time was 15 months (IQR 6–30) and participants made up a total of 4831 scheduled follow-up visits from baseline to M42 (Figure 1). During follow-up, 372 participants were LTFU (57.5%). Of those 275 participants not LTFU, 25 participants seroconverted (3.9%), one participant died (0.1%), and 249 completed follow-up (38.5%). The median follow-up time for LTFU participants was 8 months (IQR 3–16). Of the 372 LTFU participants, 122 (33%) returned to care later, meaning 250 participants were LTFU definitively. None of those who returned to care seroconverted during the period they left care or after returning to care.

Trends of LTFU according to patient’s characteristics can be found in Figure 2.

In univariate Cox analysis, risk factors for LTFU included: younger age (18–24 years old, *p* < 0.001), being followed-up in Côte d’Ivoire (*p* = 0.017), being unemployed (*p* = 0.001), being non-adherent to PrEP (no PrEP, *p* < 0.001; poor adherence, *p* = 0.003; suboptimal adherence, *p* = 0.026), both a man and a woman/more a woman/neither a woman or a man gender identity (*p* = 0.023), having a moderate to severe depression score (*p* = 0.029), having a casual partner in the previous three months (male, *p* = 0.022; female, *p* = 0.030), perceiving no HIV risk with stable male partners (*p* = 0.007) and using neither PrEP nor condoms (*p* < 0.001) or condoms only (*p* < 0.001) during most recent anal intercourse (Table 2). Protective factors for LTFU included being followed-up in Togo (*p* < 0.001), contacting PE outside of scheduled visits (*p* = 0.014), and having no casual partners (*p* = 0.006).

In the final multivariate model, participants who were LTFU were more likely to be young (18–24, adjusted hazard ratio (AHR) [95% Confidence Interval, CI], *p*-value; 1.50 [1.17–1.94], 0.002) and unemployed (1.33 [1.03–1.71], 0.027), to have a moderate to high depression score (1.63 [1.12–2.38], 0.010), to have had a casual female partner in the previous three months (1.40 [1.00–1.95], 0.047), to have had more than two male sexual partners in the previous three months (1.37 [1.03–1.83], 0.031), and to perceive no HIV risk with their stable male partner (1.61 [1.23–2.10], <0.001) (Table 2). Furthermore, participants who used neither PrEP nor condoms (2.56 [1.92–3.42], <0.001), PrEP only (1.50 [1.05–2.13], 0.025), and condoms only (2.19 [1.47–3.25], <0.001) during their most recent anal intercourse were more likely to be LTFU compared to those who used both PrEP and condoms. Meanwhile, participants who were LTFU were less likely to have contacted PE outside of scheduled visits (0.74 [0.56–0.97], 0.029). Finally, a positive trend existed between being LTFU and self-identifying as heterosexual (1.83 [0.91–3.68], 0.088).

## 4. Discussion

This analysis showed that more than half of MSM taking PrEP were LTFU for six months or more at least once in the CohMSM-PrEP study in West Africa and of them a third later returned to care, while the rest were LTFU definitely. Compared to a preliminary analysis of the cohort, definitive LTFU increased from 27% to 39% (250/647) [30]. In both the United States and in SSA, the risk of PrEP disengagement tended to increase overtime [26,27,43,44] and could explain why we found higher rates of definitive LTFU at the end of the cohort compared to the preliminary analysis.

Compared to other studies in Kenya, rates of LTFU in the present study were higher [25,26,27]. However, comparison with other studies is difficult owing to the use of different definitions of LTFU and to different follow-up times. For example, in a one-year PrEP program in Kenya, 19.1% of MSM were LTFU or stopped taking PrEP after six months, but did not define the outcome any further [25]. In two other Kenyan studies defining LTFU as being more than ninety days late for an appointment (even if they later reengaged in care), LTFU ranged from 40.3% for a median follow-up time of 4.5 months [26] to 42.5% for a median follow-up time of 5.5 months [27]. It is to be noted that, on average, LTFU participants in our study stayed in care longer than those in the Kenyan studies did (8 months median follow-up time vs. 4.5–5.5 months). Indeed, the standardization of LTFU definitions across these programs would be helpful for their evaluation and comparison. Altogether, these studies and ours highlight that LTFU is a key issue to address in PrEP programs.

In this regard, one of the most important results from the present study was the positive influence peer education had on study retention, as participants who contacted PE were less likely to be LTFU. Indeed, in CohMSM-PrEP, one of PE’s main responsibilities was to contact LTFU participants in an effort to convince them to return to care. In large part due to their persistence, a third of these participants eventually returned to care. While the literature is limited on the direct effect of peer education on retention in PrEP care, it has been shown to be useful in encouraging PrEP initiation, uptake and adherence [41,60,67,68,69,70,71,72,73,74]. In general, MSM largely prefer community-based clinics for PrEP delivery [24,75,76,77,78,79,80,81,82,83], as they offer a more comprehensive and MSM-friendly approach to PrEP provision [75]. Our present finding reinforces previous findings from CohMSM-PrEP that showed peer-based outreach over time helped reach a new profile of MSM initiating PrEP [61], that the provision of PrEP in MSM-friendly community-based clinics promoted its use, and that peer education facilitated correct PrEP adherence [60].

Although peer education facilitated study retention in the present study, vulnerability was a barrier to it, mirroring findings from previous CohMSM-PrEP studies and the literature. Financial barriers can impede PrEP users from accessing care because of cost or not having time to attend appointments, ultimately leading to disengagement with PrEP care, as findings from the US suggest [45,46,47,48,49]. In the present study, unemployed participants were more likely to be LTFU. Although participants in CohMSM-PrEP received the prevention package free of charge and were reimbursed transport costs (USD 5), this still might not have been enough in light of the hostile environment towards MSM in the region. In terms of psychosocial vulnerability, studies in the US found higher rates of LTFU among patients with mental health problems [51,53], while in South Africa and Kenya, stigmatization was associated with PrEP discontinuation [25,28]. Our findings reflect the literature, as having a moderate to severe depression score predicted LTFU. In previous CohMSM-PrEP studies, financially insecure event-driven users were less likely to have correct adherence [60] and more likely to be ineffectively protected against HIV (i.e., incorrect PrEP adherence and no condom use) [62]. Furthermore, feeling alone was associated with incorrect PrEP adherence [60]. A final risk factor related to vulnerability in the present study was younger age—A common predictor of LTFU in other studies in the US [45,47,53,55,56,57]. In general, youth MSM are particularly vulnerable to HIV infection because of the compounding effect of power imbalances related to age on existing homonegativity [84]. Indeed, one can suppose that the accumulation of several of these vulnerabilities will produce an even stronger effect on LTFU.

A final result from the present study was that HIV risk perception played a substantial role in LTFU, reflecting findings from previous CohMSM-PrEP studies and the literature that showed HIV-risk perception also facilitated PrEP initiation and engagement [50,60,61,85,86,87,88,89,90,91,92]. It is important to note that PrEP is not a lifelong tool for everyone and it is not unusual to cycle in and out of PrEP use in accordance with “seasons of risk” [93]. Some instances of PrEP disengagement are voluntary due to decreasing perceived HIV risk [45,47,48,49,50,51,52,53], such as when users enter a period of abstinence or into a monogamous relationship with an HIV-negative or a virally suppressed HIV-positive partner [54]. This was evidenced in our study’s findings with participants who did not feel at risk of HIV infection with their stable partner being more likely to leave the cohort. Participants declaring casual female partners were also more likely to leave the cohort and a positive trend existed between LTFU and self-defining as heterosexual. These findings could reflect participants feeling less “at-risk” with female partners and having less need for PrEP.

However, this disengagement based on HIV risk perception requires accuracy. Indeed, multiple studies found recent high-risk behavior before LTFU despite reporting no longer being at-risk [49,50,55,56]. Indeed this period is critical and among MSM on PrEP in SSA, seroconversions occurred primarily among MSM who had stopped taking PrEP or those who had low or no detectable drug levels in their system [30,88]. In the present study, we also found relatively recent high-risk behavior was a risk factor for LTFU, i.e., having more than two male sexual partners in the previous three months. At the time, PrEP was only available through the cohort and these participants would have been at risk of HIV infection if they continued such practices without readopting appropriate risk reduction techniques after leaving the cohort. Nonetheless, sexual behavior is dynamic and multi-partnerships in the past do not necessarily continue in the future. LTFU participants did not return to care for over six months, during which time indication for PrEP could have changed.

In general, multiple findings from CohMSM-PrEP, including those from the present study, suggest certain behavioral, psychosocial and socioeconomic factors influence multiple levels of the PrEP care cascade simultaneously. They also highlight the value of community-based methods in such care. Specifically, in terms of improving LTFU, we recommend strengthening the role of PE in PrEP care as they can provide resources to users in matters related to HIV risk, and socioeconomic and psychosocial vulnerabilities. Indeed, PE could provide extra support for vulnerable users or provide risk reduction counseling and empowerment interventions to improve risk perception. Finally, we recommend providers deliver appropriate messaging about PrEP as a tool that can be stopped and started based on HIV risk, so users simply discontinue PrEP and do not leave care completely.

Our study has limitations. First, it was carried out in MSM-friendly community-based clinics located in the countries’ economic capitals, so participants might not be representative of the local MSM community and our findings might not be generalizable to other contexts. However, the differences found in an analysis comparing ex-participants from a previous MSM cohort, CohMSM, with newly recruited participants in CohMSM-PrEP, suggested that adding PrEP to the existing prevention package helped reach a new profile of MSM—less connected to these clinics [61]. Second, participants’ responses might have been affected by social desirability bias regarding sensitive topics, though; this bias was minimized by training research assistants to administer all the questionnaires and by having repeated and regular contact between participants and assistants. Furthermore, this limit only concerns covariates as LTFU was measured objectively, based on follow-up visit dates. Finally, we considered participants who reengaged in care after the 6-month cutoff to be LTFU, which led to a higher rate of LTFU. These participants represented 33% of the LTFU participants (122/372) and feedback from study staff and physicians suggested they missed appointments because they were highly mobile (travel, work, etc.) and/or had enough PrEP and did not feel the need to attend the clinic. Indeed, these participants would be ideal candidates for 6-month PrEP dispensing with HIV self-testing, which has been shown to be non-inferior to standard of care [94]. Furthermore, this definition is useful to determine interruptions in care, especially in contexts where PrEP is not available and for future long-acting injectable forms of PrEP. Indeed, up to one year or more after stopping injections trace amounts of these drugs remain present in the bloodstream, which are not enough to protect from HIV infection and could lead to HIV seroconversion and the development of drug resistant HIV [95]. Our definition is helpful for identifying optimal candidates for long-acting PrEP with no extended interruptions in care.

Despite these limitations, this is the first study to explore risk factors for LTFU among MSM taking PrEP in West Africa and one of the very few in Sub-Saharan Africa. Informing on the state of the PrEP care cascade in West Africa and identifying a LTFU profile, will help guide PrEP delivery and rollout as more and more countries in the region add PrEP to their national AIDS programs. Even though a high rate of LTFU was found in the present study, our results showed that for MSM participants who stayed in care, PrEP was a well-adopted and complementary HIV prevention tool. Indeed, stakeholders should not be discouraged by LTFU rates, but encouraged by the decrease in HIV incidence previously found in the cohort [30] and the fact that participants who used both PrEP and condoms during their most recent anal intercourse were more likely to be retained in care in the present study.

## 5. Conclusions

Our findings show that LTFU from PrEP care in West African MSM is a major challenge to achieving HIV elimination. However, they also show that the involvement of PEs in the management of PrEP users helps to limit LTFU by providing them with adequate support. There is an urgent need for the recognition and funding of these staff.

## Figures and Tables

**Figure 1 viruses-14-02380-f001:**
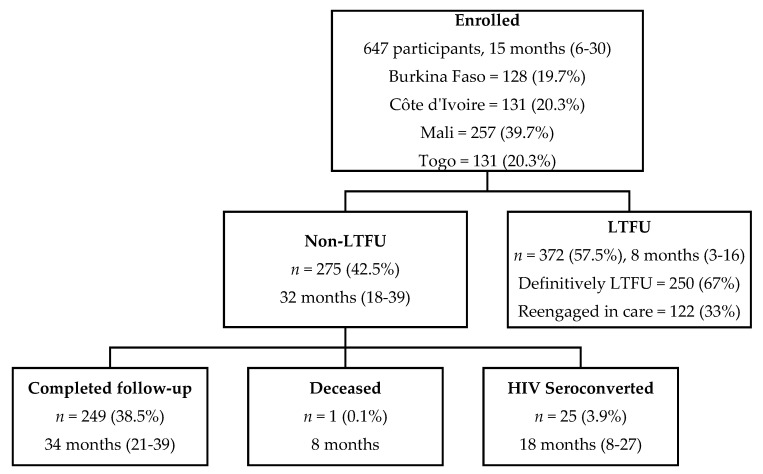
CohMSM-PrEP study population flowchart with median follow-up time in months (IQR) (Follow-up from 20 November 2017–30 June 2021; 647 participants contributed to 4831 visits from baseline to M42).

**Figure 2 viruses-14-02380-f002:**
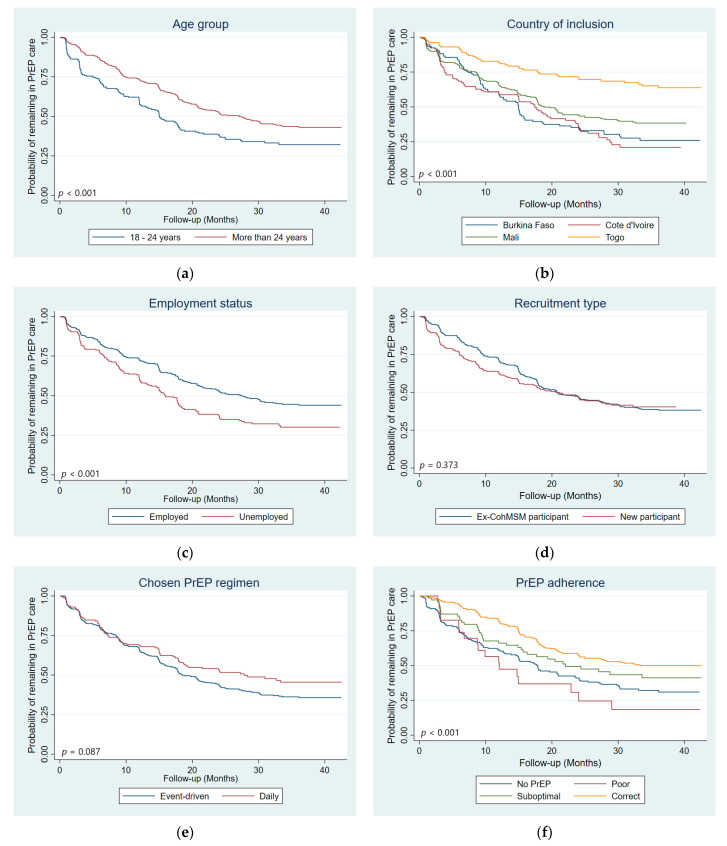
Kaplan–Meier curves with log-rank test *p*-value comparing LTFU by: (**a**) age group, (**b**) country, (**c**) employment status, (**d**) recruitment type, (**e**) chosen PrEP regimen, (**f**) PrEP adherence, (**g**) contact with peer educators outside of scheduled visits, (**h**) self-defined sexual orientation, (**i**) self-defined gender identity, (**j**) depression score, (**k**) had a casual male sexual partner in the previous three months, (**l**) had a casual female sexual partner in the previous three months, (**m**) number of male sexual partners in the previous three months, (**n**) Combined prevention use during most recent anal intercourse, (**o**) HIV risk perception with stable male partner, and (**p**) HIV risk perception with casual male partners (*n* = 647 participants, 4831 visits).

**Table 1 viruses-14-02380-t001:** Characteristics of study participants at baseline, *n* = 647.

Variable	*n* (%)
Age group (in years) ^1^	
18–24	314 (48.5)
>24	275 (42.5)
Country of inclusion	
Burkina Faso	128 (19.7)
Cote d’Ivoire	131 (20.3)
Mali	257 (39.7)
Togo	131 (20.3)
Employment status ^2^	
Employed	283 (43.7)
Unemployed	299 (46.2)
Recruitment type	
Ex-CohMSM participant	322 (49.8)
New participant	325 (50.2)
Chosen PrEP regimen	
Event-driven	451 (73.1)
Daily	164 (26.6)
PrEP adherence ^3,^*	
No PrEP	111 (21.3)
Poor	27 (5.2)
Suboptimal	84 (16.1)
Optimal	213 (40.8)
Contacted PE outside of scheduled visits ^4^	
Yes	162 (25.0)
No	405 (62.6)
Self-defined sexual orientation ^5^	
Heterosexual	15 (2.3)
Homosexual/gay	228 (35.2)
Bisexual	334 (51.6)
Self-defined gender identity ^6^	
Both a man and a woman; more a woman; neither a woman or a man	229 (35.4)
Man or boy	354 (54.7)
Depression score (PHQ-9)	
Moderate to high	295 (45.6)
Minimal	352 (54.4)
Had a casual male sexual partner in the previous three months ^7^	
Yes	379 (58.6)
No	261 (40.3)
Had a casual female sexual partner in the previous three months ^8^	
Yes	131 (20.3)
No	510 (78.8)
Number of male sexual partners in the previous three months ^7^	
≤2	416 (64.3)
>2	224 (34.6)
HIV risk perception with stable male partner ^9^	
No risk	192 (29.7)
At risk	265 (41.0)
No stable partner	181 (28.0)
HIV risk perception with casual male partners ^10^	
No risk	96 (14.8)
At risk	281 (43.4)
No casual partners	260 (40.2)
Combined prevention use during most recent anal intercourse ^11,^*	
No PrEP & no condom use	139 (26.7)
PrEP use only	149 (28.6)
Condom use only	50 (9.6)
PrEP & condom use	172 (33)

* PrEP-related outcomes were only available from the M3 follow-up questionnaire since there was no pill intake at M0. ^1^ Range = 18–57, mean age (SD) = 25.3(5.8), 58 (9.0%) missing values. ^2^ 65 (10.1%) missing values. ^3^ 86 (16.5%) missing values. ^4^ 80 (12.4%) missing values. ^5^ 70 (10.8%) missing values. ^6^ 64 (9.9%) missing values. ^7^ 7 (1.1%) missing values. ^8^ 6 (0.9%) missing values. ^9^ 9 (1.4%) missing values. ^10^ 10 (1.6%) missing values. ^11^ 11 (2.1%) missing values. SD: standard deviation; CohMSM: cohort of MSM; PrEP: pre-exposure prophylaxis; PE: peer educator; PHQ-9: Patient Health Questionnaire-9.

**Table 2 viruses-14-02380-t002:** Determinants of loss to follow-up from PrEP care (*n* = 647 participants, 4831 measures) ^1^.

			Univariate	Multivariate
Variable	*n* (%) ^2^	Mean Follow-Up Time in Months (SD)	HR [95% CI], *p*-Value	aHR [95% CI], *p*-Value
Age group (in years)				
18–24	759 (35.9)	7.9 (7.7)	1.64 [1.31–2.05], <0.001	1.50 [1.17–1.94], 0.002
>24	1253 (59.4)	12 (9.1)	1 (ref)	1 (ref)
Employment status				
Employed	1137 (53.9)	11.4 (9.3)	1 (ref)	1 (ref)
Unemployed	863 (40.9)	9.0 (8.0)	1.44 [1.5–1.8], 0.001	1.33 [1.03–1.71], 0.027
Chosen PrEP regimen				
Event-driven	1356 (64.3)	11.3 (8.5)	0.82 [0.64–1.06], 0.125	
Daily	370 (17.5)	10.3 (8.6)	1 (ref)	
PrEP adherence				
No PrEP	561 (26.6)	10.8 (8.9)	1.89 [1.44–2.49], <0.001	
Poor	114 (5.4)	11.0 (7.9)	2.22 [1.32–3.73], 0.003	
Suboptimal	205 (9.7)	11.6 (8.3)	1.53 [1.05–2.23], 0.026	
Optimal	751 (35.6)	14.2 (7.6)	1 (ref)	
Contacted PE outside of scheduled visits				
Yes	601 (28.4)	11.4 (9.7)	0.73 [0.56–0.94], 0.014	0.74 [0.57–0.97], 0.032
No	1349 (63.9)	9.9 (8.4)	1 (ref)	1 (ref)
Self-defined sexual orientation				
Heterosexual	43 (2.0)	9.0 (5.5)	1.9 [0.96–3.75], 0.066	1.84 [0.92–3.69], 0.086
Homosexual/gay	730 (34.6)	9.8 (8.8)	1.18 [0.93–1.49], 0.170	1.20 [0.94–1.53], 0.149
Bisexual	1210 (57.4)	10.8 (9.1)	1 (ref)	1 (ref)
Self-defined gender identity				
Both a man and a woman; more a woman; neither a woman or a man	784 (37.2)	9.9 (8.2)	1.29 [1.04–1.61], 0.023	
Man or boy	1216 (57.6)	10.5 (9.2)	1 (ref)	
Depression score (PHQ-9)				
Moderate to high	954 (45.2)	9.4 (8.5)	1.44 [1.04–1.98], 0.029	1.63 [1.12–2.38], 0.010
Minimal	1156 (54.8)	10.7 (8.9)	1 (ref)	1 (ref)
Had a casual male sexual partner in the previous three months				
Yes	1058 (50.1)	9.3 (7.8)	1.33 [1.04–1.69], 0.022	
No	1058 (49.7)	10.9 (9.6)	1 (ref)	
Had a casual female sexual partner in the previous three months				
Yes	345 (16.4)	8.7 (8.9)	1.40 [1.03–1.90] 0.030	1.40 [1.00–1.95], 0.047
No	1762 (83.5)	10.4 (8.7)	1 (ref)	1 (ref)
Number of male sexual partners in the previous three months				
≤2	1497 (70.9)	10.3 (9.2)	1 (ref)	1 (ref)
>2	609 (28.9)	9.9 (7.7)	1.26 [0.99–1.61], 0.058	1.37 [1.03–1.83], 0.031
HIV risk perception with stable male partner				
No risk	760 (36.0)	10.2 (9.1)	1.41 [1.10–1.81], 0.007	1.61 [1.23–2.10], <0.001
At risk	790 (35.4)		1 (ref)	1 (ref)
No stable partner	597 (28.3)	9.1 (7.7)	1.04 [0.78–1.39], 0.766	1.04 [0.73–1.46], 0.843
HIV risk perception with casual male partners				
No risk	321 (15.2)	8.6 (7.6)	1.08 [0.78–1.49], 0.639	
At risk	728 (34.5)	9.7(7.8)	1 (ref)	
No casual partners	1055 (50.0)	10.9 (9.5)	0.79 [0.60–1.03], 0.083	
Combined prevention use during most recent anal intercourse				
No PrEP & no condom use	801 (37.9)	8.35 (9.2)	2.25 [1.71–2.96], <0.001	2.56 [1.92–3.42], <0.001
PrEP use only	367 (17.4)	13.1 (7.2)	1.32 [0.95–1.84], 0.100	1.50 [1.05–2.13], 0.025
Condom use only	267 (12.7)	5.6 (6.8)	2.21 [1.52–3.23], <0.001	2.19 [1.47–3.25], <0.001
PrEP & condom use	635 (30.1)	13.7 (8.1)	1 (ref)	1 (ref)

^1^ Time-varying Cox’s proportional hazards model adjusted for country-fixed effects and recruitment type, AIC = 3507. ^2^ Visits from non-LTFU participants accounted for 2721 measures; visits from LTFU participants accounted for 2110 measures; the difference between the total sample and the total number of measures for each variable corresponds to missing values. SD: standard deviation; HR: hazard ratio; aHR: adjusted hazard ratio; CI: confidence interval; ref: reference; CohMSM: cohort of men who have sex with men; PrEP: pre-exposure prophylaxis; PE: peer educator; PHQ-9: Patient Health Questionnaire-9.

## Data Availability

Data can be requested by submitting a study proposal to the scientific board of the CohMSM-PrEP project (christian.laurent@ird.fr). Proposals will be evaluated for compatibility with the CohMSM-PrEP project and overlap with ongoing work. The codes generated during and/or analyzed during the current study are available from the corresponding author on request.

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
