# Peer review of "Loss to Follow-Up from HIV Pre-Exposure Prophylaxis Care in Men Who Have Sex with Men in West Africa"

_viruses, 2022, doi:10.3390/v14112380_

Round 1

Reviewer 1 Report

This is a well-written and thoughtful paper from a distinguished team describing the loss to follow-up among men prescribed TDF/FTC HIV PrEP in clinics located in four West African nations. 

There are limited data regarding the trajectories of PrEP use among people in West Africa, including the key population of men who have sex with men. The authors expand on their prior work to examine the rate of loss to follow-up, defined as failure to return to clinic within six months of their last clinic visit, as well as the covariates associated with this outcome. The paper provides important observations regarding factors that may contribute to PrEP persistence in this population and setting. 

Strengths of the study and manuscript include a large sample size of 647 participants, inclusion of four different countries to enhance generalizability, and use of multivariable analyses to find independent associations with loss to follow-up. 

A weakness is a lack of clarity regarding what data timepoints are contributing to the findings. Baseline and, for sexual- and PrEP-related behaviors, month 3 data are presented. There is no mention of any subsequent survey data including those from visits more proximal to the last clinic visit. For those lost to follow-up, some of the covariates most of interest can be dynamic - such as risk behaviors, relationship status, and depression and therefore can change since month 3.  As mentioned, PrEP and risk alignment and misalignment are critical to understanding PrEP adherence. Cessation of PrEP when there is no risk of exposure to HIV is not consequential, while lapses in TDF/FTC during periods of risk is. Few expect PrEP to be taken forever, so at some point all PrEP users will be "lost to follow-up".  We need to understand better if here the failure to return was justified or tragic. 

The authors also acknowledge that their definition of loss to follow-up includes those who return to PrEP.  This was a third of those "lost to follow-up" - at least transiently. More about those who returned would be helpful. 

There can also more information provided about seroconversion. Besides the 25 people who seroconverted in the non-LTFU group, were any who were lost to follow-up diagnosed with HIV infection, for example on return to care? 

Lastly, the authors raise an important point regarding risk perception. It seems unclear if the low risk perception of some respondents is a true reflection of their risk or an under-appreciation of their potential for infection. The authors appear to be unsure themselves and one wonders if there are sexual behavior data that can help make more evident the actual risk these men faced - beyond month 3 (see above).

Overall, this is an interesting and important paper that reveals potential limitations of even well-designed PrEP programs in West Africa, and points the way toward enhanced PrEP engagement strategies. Additional details from the study will strengthen the paper and message. 

Author Response

Please see the attachment for our Point-by-point response. We also thoroughly reviewed the manuscript for spelling and grammar mistakes. 

Reviewer 2 Report

The outcome, LTFU, as currently defined is problematic and warrants further explanation, which may impact the analyses. LTFU was defined as the first episode of LTFU if they did not return to the clinic six months after their last visit, regardless if they re-engaged later in care. How many participants met your LTFU definition, but then later re-engaged in care? In general, the authors need to provide a justification for why they chose to define LTFU is such a way, given a participant could re-engage in care later and would therefore, not technically be lost to LTFU. 

In the Discussion, the authors mention LTFU was defined differently in other studies, but do not elaborate in what ways. Elaborate and describe in what ways LTFU could be better measured and defined to allow comparisons across different studies. 

Feeling "less at risk" differs for actual risk. "Seasons of risk" makes sense as it reflects the reality, yet at the same time, could negatively impact LTFU when in fact participants could benefit by remaining on PrEP. The authors ought to improve this section by detailing ways to help prevent LTFU from occurring - something that's largely not well addressed in detail in the Discussion. Provide examples, be specific to help improve prevention science.

Author Response

(The authors gave the same response as above.)
